# Intralocus conflicts associated with a supergene

Lina M. Giraldo-Deck [1✉], Jasmine L. Loveland[1], Wolfgang Goymann [2], Barbara Tschirren [3], Terry Burke [4], Bart Kempenaers [5], David B. Lank [6✉] & Clemens Küpper [1✉]

Chromosomal inversions frequently underlie major phenotypic variation maintained by divergent selection within and between sexes. Here we examine whether and how intralocus conflicts contribute to balancing selection stabilizing an autosomal inversion polymorphism in the ruff *Calidris pugnax*. In this lekking shorebird, three male mating morphs (Independents, Satellites and Faeders) are controlled by an inversion-based supergene. We show that in a captive population, Faeder females, who are smaller and whose inversion haplotype has not undergone recombination, have lower average reproductive success in terms of laying rate, egg size, and offspring survival than Independent females, who lack the inversion. Satellite females, who carry a recombined inversion haplotype and have intermediate body size, more closely resemble Independent than Faeder females in reproductive performance. We inferred that the lower reproductive output of Faeder females is most likely balanced by higher than average reproductive success of individual Faeder males. These findings suggest that intralocus conflicts may play a major role in the evolution and maintenance of supergene variants.

[1] Research Group Behavioural Genetics and Evolutionary Ecology, Max Planck Institute for Ornithology, Eberhard-Gwinner-Str., 82319 Seewiesen, Germany. [2] Department of Behavioural Neurobiology, Max Planck Institute for Ornithology, Eberhard-Gwinner-Str., 82319 Seewiesen, Germany. [3] Centre for Ecology and Conservation, University of Exeter, Penryn TR10 9FE, UK. [4] Ecology and Evolutionary Biology, School of Biosciences, University of Sheffield, Sheffield S10 2TN, UK. [5] Department of Behavioural Ecology and Evolutionary Genetics, Max Planck Institute for Ornithology, Eberhard-Gwinner-Str., 82319 Seewiesen, Germany. [6] Department of Biological Sciences, Simon Fraser University, Burnaby, BC V5A 1S6, Canada. ✉email: lgiraldo@orn.mpg.de; dlank@sfu.ca; ckuepper@orn.mpg.de

Antagonistic pleiotropy, where alleles increase certain fitness components at the detriment of others, can result in genetic trade-offs for fitness traits[1–5]. It is widespread for sexually antagonistic loci, which underlie the expression of reproductive traits in males and females. At these loci, intralocus sexual conflict (IASC) arises as the sexes have different phenotypic optima because of divergent selective pressures[4,6,7]. IASCs can lead to balancing selection, particularly in combination with negative frequency-dependent selection[8,9], or when strong directional selection operates on each sex[10]. Intralocus conflicts can also occur between morphs, because similar to the sexes, morphs experience different selective pressures, but share largely the same genome[11,12].

Chromosomal inversions can give rise to supergenes that maintain different adaptive allele combinations through suppression of recombination[13–16]. Similar to sex chromosomes, these supergenes can thus provide the genetic substrate for divergent selection, increasing the phenotypic diversity within species. Inversion-based supergenes frequently underlie morphs with alternative reproductive phenotypes[5,17–19] and can help to resolve conflicts between reproductive tactics[11]. However, when located on autosomes, supergenes may increase the potential for IASCs, because the different morph-sex classes will typically have non-overlapping phenotypic optima. Intralocus conflicts between sexes and morphs may thus play an important role in stabilizing inversion polymorphisms. Supergene variants are thought to be maintained by balancing selection, mediated by negative frequency-dependent selection, spatially variable selection, selfish genetic elements, overdominance and life-history trade-offs, although the exact mechanisms have rarely been fully characterized[5,13,20–26].

Here we examine sexual antagonism as a source of balancing selection maintaining an autosomal inversion polymorphism in ruffs (*Calidris pugnax*). Ruffs feature three well-described male mating morphs that differ in aggressive and courtship behavior, body size, circulating hormone levels, and relative testis size:[17,27] (1) large territorial Independents gather on leks and compete aggressively for visiting females, (2) diminutive Faeders sneak copulations through female mimicry, and (3) semi-cooperative, intermediate-sized Satellites display on leks with Independents. The differences between morphs are encoded by an autosomal inversion region with dominant inversion alleles[17,18]. The original inversion arose approximately 3.8mya ago and resulted in the Faeder morph, whereas the Satellite allele arose later through rare recombination events between the ancestral arrangement and the Faeder allele[18]. Independents are homozygous for the ancestral arrangement, whereas Faeders and Satellites are heterozygous inversion carriers, since the inversion alleles are homozygous lethal[17]. Because the inversion is autosomal, the three morphs occur in both sexes. The lethality of homozygotes also implies substantial fitness costs from matings between inversion morphs, which are particularly borne by the inversion females, who provide all parental care[28].

Although the supergene variants have the most flamboyant consequences for male phenotypes, they may also have fitness consequences for females. For each morph, females are approximately 40% smaller than males, resulting in particularly small Faeder females[29,30]. Here, we investigate variation in female reproductive success in relation to the different supergene variants. Based on these results, we explore potential mechanisms keeping the inversion polymorphism stable. We found that compared to Independent females, Faeder females had lower reproductive success. We identified disproportionally higher fertilization success by Faeder males as the most likely mechanism to maintain the Faeder haplotype in the population.

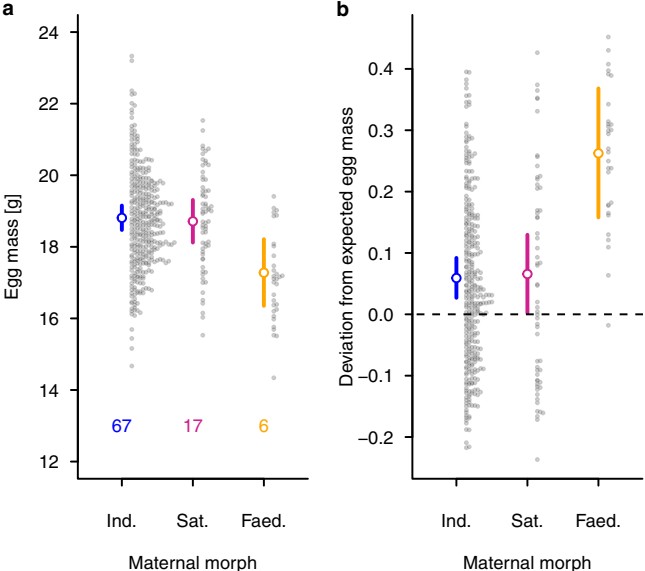

**Fig. 1 Variation in egg mass by female morph. a** Egg mass according to female morph when statistically controlled for female ID and sampling year. The probabilities of Faeder females ($N_{Eggs}$ = 32, $N_{Females}$ = 6) laying lighter eggs than Independent ($N_{Eggs}$ = 376, $N_{Females}$ = 67) or Satellite females ($N_{Eggs}$ = 66, $N_{Females}$ = 17) were higher than 0.99, differences between Independent and Satellite females were not statistically clear. **b** Deviation of the observed egg mass from the egg mass expected according to female body mass ([observed egg mass/expected egg mass]-1) controlling for female ID and sampling year. Zero indicates the expected egg mass[41]. The probabilities of Faeder females ($N_{Eggs}$ = 30, $N_{Females}$ = 5) having a higher deviation than Independent ($N_{Eggs}$ = 345, $N_{Females}$ = 58) or Satellite ($N_{Eggs}$ = 59, $N_{Females}$ = 14) females were both higher than 0.99. Given are means and 95% CrI for Independents in blue, Satellites in purple and Faeders in orange. Full model details are given in Supplementary Tables 2 and 3. Source data are provided as a source data file.

## Results

**Morph-dependent variation in female reproductive output**. We monitored the reproductive investment and success of 186 female ruffs (118 Independents, 48 Satellites and 20 Faeders) over 3 years. We established morph- and age-specific laying rates from breeding pens containing females of a given morph and age class. Faeder females had a lower laying rate than Independent or Satellite females (Supplementary Fig. 1, Supplementary Table 1). Faeder females also produced smaller eggs (mean [95% credible interval (CrI)]: 17.2 g [16.8g–17.6 g]) than Independent (18. 8 g [18.6g–18.9 g]) or Satellite females (18.6 g [18.4g–18.8 g]), (Fig. 1a, Supplementary Table 2). However, relative to their body mass Faeder females produced substantially larger eggs than Independents or Satellites (Fig. 1b, Supplementary Table 3).

Circulating steroid concentration during the reproductive period is a major physiological difference between male morphs induced by the supergene variants[17,31]. Similarly, endocrinal variation may also exist in females. Consistent with the differences reported in males[31], yolks of Faeder females contained higher concentrations of androstenedione than yolks of the other morphs (Supplementary Fig. 2). However, the eggs did not differ clearly across morphs in yolk testosterone or progesterone concentrations (Supplementary Fig. 2).

Because all eggs and offspring experienced standardized incubation and rearing conditions, survival differences between young ruffs of the same sex and morph must have largely been caused by variation in egg mass and/or composition. To disentangle these two effects, we statistically controlled for egg mass in models comparing hatching and fledging success. This

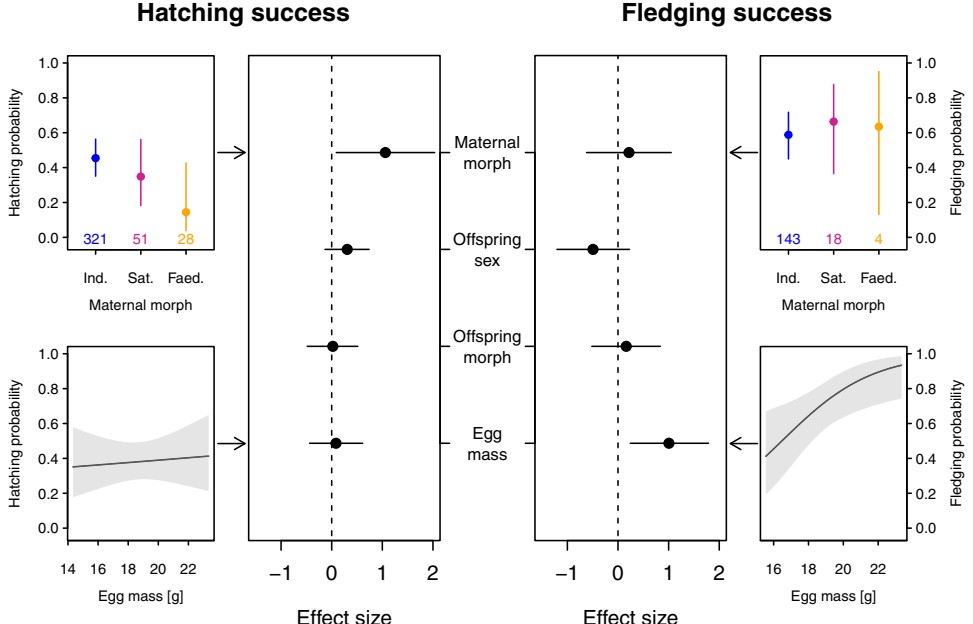

**Fig. 2 Co-variates of hatching and fledging success.** Central panels: Effect sizes (means ± 95% CrI) of maternal morph (controlled for egg mass), offspring sex, offspring morph and egg mass on hatching and fledging success. The models include mother ID and sampling year as random factors. Side panels: Influence of the maternal effects: maternal morph (top) and egg mass (bottom) on hatching and fledging probabilities (means ± 95% CrI). For visualization, the estimates of maternal morph on hatching and fledging probabilities were calculated for Independent male embryos using the overall mean egg mass (hatching: 18.61 g, fledging: 18.68 g). Similarly, the estimates of egg mass on hatching and fledging probabilities were visualized using Independent mothers and Independent female offspring. For hatching success, the probabilities of effect sizes being less than zero (P(β < 0)) were: 0.02 for maternal morph, 0.09 for offspring sex, 0.47 for offspring morph, and 0.38 for egg mass. For fledging success, the probabilities of effect sizes being less than zero (P(β < 0)) were: 0.31 for maternal morph, 0.91 for offspring sex, 0.32 for offspring morph, and <0.01 for egg mass. Sample sizes were 400 eggs from 87 mothers for hatching success and 165 chicks from 62 mothers for fledging success. Sample size details are provided in methods and model details and all other estimates on hatching and fledging probabilities are provided in Supplementary Tables 4 and 5. Source data are provided as a Source Data file.

means that the effects of maternal morph are disentangled from morph-specific egg mass and are thus most likely caused by variation in egg composition. After controlling for egg mass, maternal morph had the strongest effect on hatching success among all predictors (Fig. 2, Supplementary Table 4), with eggs laid by Independent females having the highest hatching success, followed by those of Satellite females and then Faeder females (Fig. 2, Supplementary Fig. 3, Supplementary Table 5). Eggs from Faeder females experienced substantially higher mortality between 5 and 8 days of embryonic development (Supplementary Fig. 4). Egg mass itself had no discernible effect on hatching success, but a strong impact on offspring survival after hatching (Fig. 2, Supplementary Fig. 3, Supplementary Table 4). Young hatching from heavier eggs survived better and were more likely to reach fledging age than young hatched from lighter eggs (Fig. 2, Supplementary Fig. 3, Supplementary Table 4). Male ruff chicks were particularly vulnerable when hatching from small eggs, whereas egg size did not seem to overly affect survival of female chicks (Supplementary Fig. 5).

Overall, Faeder females produced substantially fewer fledglings (3.0 fledglings assuming an Independent female would lay 100 eggs [0.5–15.8 fledglings]) than Independent (24.7 fledglings [16.7–35.2 fledglings]) and Satellite females (16.3 fledglings [6.7–34.1 fledglings]) (Fig. 3a).

**Sexual antagonism and the maintenance of the inversion polymorphism.** Given that ruffs are a cosmopolitan, panmictic species[32] and that in wild populations morph frequencies appear generally stable over time and space[28], the lower annual reproductive success of Faeder females must be compensated for by

higher reproductive success by Faeder males and/or higher adult survival of this morph. We used a set of analytical evolutionary models to identify the main mechanism for this compensation. We first estimated the fertilization success of male morphs necessary to keep allelic frequencies stable over time. For this model, we used sex-specific adult morph ratios obtained from the wild with genetic markers[18], assumed no differences in survival between morphs and no morph-assortative mating, but took into account the homozygous lethality of the inversion. Based on the observed reproductive output by captive females, Independent males would need to fertilize 76% [73–81%] of the eggs, Satellite males 22% [17–25%] and Faeder males 2.4% [1.5%–2.6%] to keep allele frequencies stable (Table 1). Given the morph frequencies of males[18] these proportions translated into a required per capita fertilization success of 0.94 [0.90–1.00] for Independent males, 1.20 [0.93–1.35] for Satellite males and 2.94 [1.91–3.10] for Faeder males (Fig. 3b). The required male reproductive success of Faeders was similar in an alternative model where we assumed that Faeder females did not reproduce at all (Fig. 3b). At the opposite extreme, full compensation through morph-biased survivorship alone seems unrealistic, as individual Faeders would need to survive substantially longer (females: 8.2 [1.6–53.4] times the Independent lifespan; males: 3.1 [2.0–3.4] times the Independent lifespan). The importance of a relatively high fertilization success of Faeder males required for the persistence of the Faeder haplotype becomes even more evident when considering the combination of reduced reproductive output by Faeder females and very low frequencies of the Faeder morph. According to our model, both Satellite and Faeder females contribute little to the maintenance of ruff populations and even to the persistence of the inversion morphs. The vast majority of all fledglings (94%

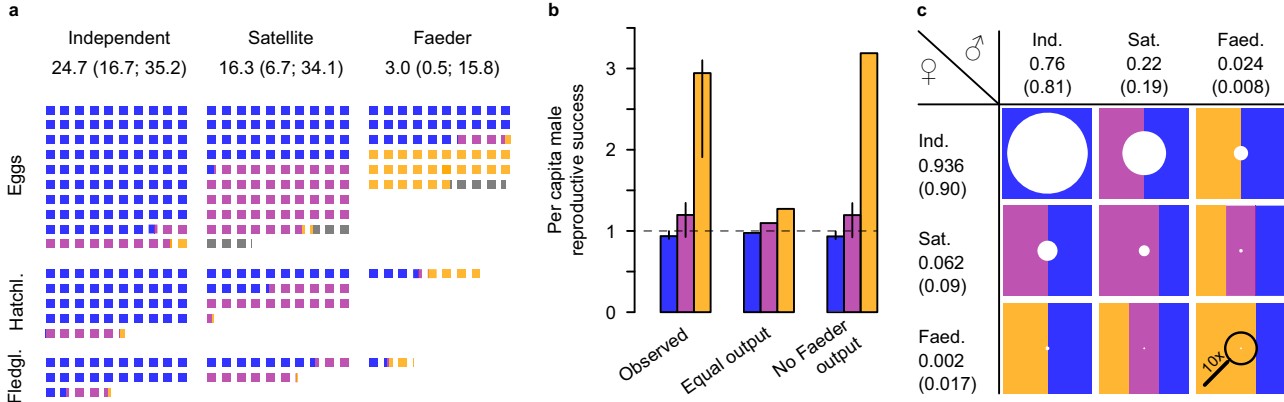

**Fig. 3 Fertility rates and reproductive success by morph and sex. a** Relative reproductive output for each female morph. Each square represents an offspring and we depict morph output relative to scores of a hundred eggs laid by Independent females. Square color represents the expected offspring morph based on the "Observed" model prediction for a stable polymorphism. Numbers indicate the mean (95% CrI) number of fledglings an individual female would produce. Independents are represented in blue, Satellites in purple, Faeders in orange and homozygous lethal eggs in gray. **b** Inferred male reproductive success (means and 95% CrI) needed to keep allelic frequencies stable according to three different scenarios of female reproductive output: observed differences in reproductive output for females in captivity ("Observed"), no differences in reproductive output between morphs ("Equal output"), and assuming that Faeder females do not produce any fledglings ("No Faeder output"). Horizontal broken line refers to equal mean reproductive success across morphs. The models use the posterior distributions ($N = 10,000$ simulated values) of annual laying rate, hatching and fledging success. Source data are provided as a Source Data file. **c** Inferred sex-morph contribution to the next fledging cohort. The row numbers indicate the proportion of fledglings in a population produced by each female morph, and column numbers refer to the proportion of fledglings produced per male morph. In parentheses, the sex-specific adult morph ratios obtained from the wild (calculated from Supplementary Table 3 in ref. [18]) are given. Each circle area represents the specific fledging proportion that each sex-morph combination contributes to the next cohort with the sum of all circle areas being 1. The colored areas indicate the morph proportions of the fledglings for each cross.

**Table 1 Proportions of eggs fertilized by male morphs required to keep allelic frequencies stable according to three different models.**

| model | Independent | Satellite | Faeder |
|---|---|---|---|
| Observed | 0.76 | 0.22 | 0.024 |
| | (0.73; 0.81) | (0.17; 0.25) | (0.015; 0.026) |
| Equal output | 0.79 | 0.20 | 0.010 |
| No | 0.75 | 0.22 | 0.026 |
| Faeder output | (0.73; 0.80) | (0.17; 0.25) | |

The "Observed" model considers the observed differences in female reproductive output between morphs. The "Equal output" model does not consider the differences in reproductive output between female morphs. The third model assumes that Faeder females do not reproduce ("No Faeder output").

[88–97%]) had Independent mothers (Fig. 3c). Even inversion fledglings were mainly produced by Independent females, which would raise 88% [57–96%] of all Faeders and 70% [56–90%] of all Satellites surviving until fledging (Fig. 3c). Thus, the inversion alleles are largely maintained in the population through matings by male carriers of these alleles with Independent females.

## Discussion
Chromosomal inversions can result in adaptive genetic changes, enabling their spread in populations[14,20,33]. Most persistent inversion alleles do not reach fixation, but are maintained at stable frequencies by balancing selection[34]. Here, we show that in ruffs, the dominant inversion alleles that underlie male alternative reproductive tactics have detrimental effects on reproductive success in females, and are most likely maintained by higher-than-average reproductive success in males. Faeder females laid fewer and smaller eggs with reduced offspring survival compared to Independent or Satellite females. For most fitness components, Satellite females underperformed Independent females, but the difference in reproductive output between Satellite and Independent females was not statistically clear because of wide

credible intervals. In addition to the reproductive components measured in our study, reproductive success in both inversion morphs will be affected by the homozygous lethality of the inversion, barring strongly non-assortative mating. Satellite females may be unable to fully compensate for this handicap, given that they had intermediate mean values for all measures of reproductive success (Supplementary Tables 1, 2, and 4).

Faeder females are distinctively smaller than Independent or Satellite females[29,30]. Their small body size could have constrained egg production, as egg size, a polygenic trait with high heritability[35–38] is typically positively correlated with female body size[35,38–41]. Although Faeder females produced smaller eggs than Independent or Satellite females, they produced the largest eggs relative to their body size. The large relative egg size could imply a higher cost of egg production[39], whereby egg size may trade off with number of laid eggs. The observed lower laying rate in Faeder females could be either the result of individual females producing fewer eggs per season or, alternatively, being more likely to skip reproduction in some years.

Our results suggest that the production of relatively large eggs remains an evolutionary stable strategy, despite the implied costs for Faeder females, because egg size affected chick survival (Fig. 2). This is consistent with previous findings in shorebirds with pronounced sexual size dimorphism[41–43]. Male ruff chicks were particularly vulnerable when hatching from small eggs, whereas egg size did not seem to overly affect survival of female chicks (Supplementary Fig. 5). The survival differences may point to different selective pressures on the sexes during development[29]. In birds, females of species with pronounced male-biased sexual size dimorphism have been observed to lay larger eggs than expected based on their size[44,45]. This may provide an adaptive mechanism to increase the survival of their male offspring, given that males require more energy for growth and maintenance of their larger body mass[46].

Embryos from Faeder mothers had a higher mortality than embryos from Independent mothers regardless of embryonic morph or egg size. The survival differences were most likely caused by variation in egg composition between morphs, as all

eggs were incubated under standardized conditions. The analysis of yolk steroid hormones revealed that Faeder females had higher androstenedione concentrations than Independent or Satellite females. The exact mechanisms of steroid allocation into the yolk and the impact of androstenedione on the embryonic physiology remain unknown. However, the increased androstenedione concentrations of Faeder eggs could point to currently unknown adaptive compensatory mechanisms for the reduced reproductive output in Faeder females. In collard flycatcher *Ficedula albicollis* high yolk androstenedione concentrations are associated with higher female fitness[47] and higher offspring survival[47,48]. Alternatively, the high androstenedione levels in egg yolks produced by Faeder females could be the result of a genetic correlation for androstenedione in males and females, as Faeder males have high circulating androstenedione concentrations[17,31]. We conclude that supergene variants directly affected female reproduction, although the proximate consequences of the supergene variation for female reproduction, physiology and morphology requires further studies.

Supergenes have been proposed as adaptive solutions to help resolve intralocus conflicts[11]. In the ruff, the inversion arrangements may have reduced intralocus conflict among males pursuing different mating strategies by capturing alleles that increase the mating success of Faeder males. But at the same time, the inversion allele altered the expression of female reproductive traits away from their phenotypic optima. Degeneration of supergene variants through accumulation of mutational load, which is frequently observed at inversion haplotypes[49–52] may have contributed to the escalation of this IASC. The degeneration may have compromised genes that are vital for female reproductive physiology, affecting unknown metabolic components of the eggs that are crucial for embryonic survival. However, for Faeder males, loss-of-function mutations that affect conspicuous ornaments and/or display behavior, plus smaller body size, may have been beneficial and enabled a more effective "sneaking" strategy. It has been argued that IASC is likely to be greater for tactics where sexes are most dimorphic[53]. Yet, our results on reproductive fitness in male and female Faeders suggest that IASC is most pronounced in the morph where males and females appear phenotypically most similar during the lekking season. The antagonistic effects of the Faeder allele rather support the hypothesis that adaptation in rare morphs is constrained through the correlated response to selection in the common morphs[11,12]. In Satellites, which occur at higher frequencies than Faeders in natural populations[18], IASCs appear to be less escalated. The Satellite allele is the result of recombination between Faeder and ancestral Independent alleles[17,18]. Genetic modification of inversion alleles may re-instate fitness lost through previous sequence degeneration[44]. Consistent with this, the recombined Satellite allele had a lower impact on female reproductive success than the Faeder allele. Further genomic modifications, such as a tight linkage with the sex chromosomes[12], could reduce the IASC even more.

The different selection pressures acting on males and females are key to understanding the maintenance of alternative reproductive tactics[54], with sexual conflict playing an important role in the persistence of behavioral polymorphisms[54,55]. Sexual antagonism in interaction with negative frequency-dependent selection may contribute to balancing selection[8,9], particularly in systems with strong sexual selection[10]. Although in ruffs, Faeder females may contribute little to maintain the Faeder allele in the population (Fig. 3c), the physical location of the inversion on an autosome sustains Faeder females despite their poor reproductive success. The resulting demographic deficit would be offset if negative frequency-dependent selection increases the mating success of individual Faeder males the rarer their morph becomes

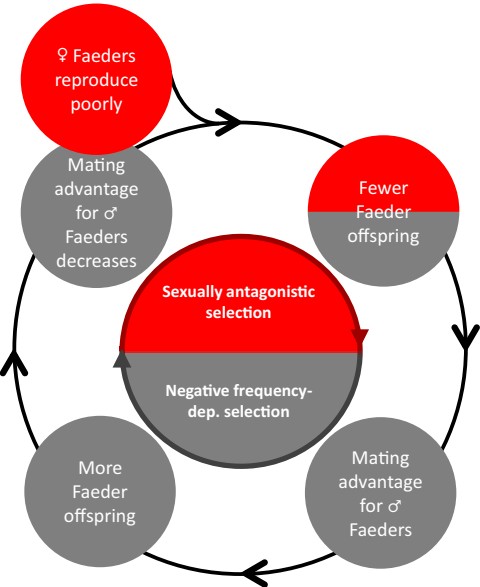

**Fig. 4 Dynamic relationship between sexual antagonism and negative frequency-dependent selection represents a possible mechanism to stabilize the frequency of the Faeder allele in ruffs.** As Faeder females have low reproductive success and produce few Faeder fledglings, Faeder males benefit through negative frequency-dependent selection of their alternative reproductive tactic and produce more Faeder offspring. The spread of the Faeder allele will be limited by its sexually antagonistic effects on female reproductive output and negative frequency-dependent selection reducing the reproductive output of Faeder males when they become more common.

(Fig. 4). Similarly, negative frequency-dependent selection in combination with mating advantages has been shown to favor inversion haplotypes that have accumulated recessive mutational load in *Heliconius* butterflies[50].

The ruff inversion, specifically the Faeder allele, has hallmarks of a parasitic genetic element that is unable to persist without the ancestral arrangement. First, during initial development, the inversion haplotypes always require a partner chromosome of the ancestral haplotype, since the breakpoint disruption of *CENPN* pertinent to Faeder and Satellite alleles are homozygous lethal[17,18]. Second, the two male mating strategies associated with the inversion[56] largely feed off the courtship efforts of Independent males, as females are attracted to the leks created by Independents and most matings occur on the leks[57]. Third, as we show here, to persist in the population the inversion alleles heavily rely on the maternal investment and maternal care efforts of Independent females. These females raise disproportionately large shares of new Satellites and Faeders, whereas particularly Faeder females contribute little to maintain the inversion variants within the population, as their own reproductive success is low (Fig. 3c).

## Methods

**Captive breeding**. We studied reproductive success of captive breeding ruff females from 2017 to 2019 at Simon Fraser University in Burnaby, British Columbia, Canada. The population consisted of approximately 300 adult individuals and was established from eggs collected near Oulu, Finland in 1985, 1989, and 1990[30]. It contains individuals from all three morphs that have been interbred for multiple generations. During the breeding season from April to end of July, females were held in breeding pens separated by morph and grouped by age, with $13.4 \pm 5.7$ (mean ± SD) females per pen (range: 7–18 females). We assigned female groups to different pen locations every year resulting in 33 different pen replicates (Supplementary Fig. 1, $N_{Independents} = 19$, $N_{Satellites} = 11$, $N_{Faeders} = 3$). Number of females per pen had a negligible effect on our response variables (laying rate: $-0.02$ [$-0.10$; 0.05], $P(\beta < 0) = 0.75$, egg mass: 0.001 [$-0.02$; 0.20], $P(\beta < 0) = 0.45$, hatching success: 0.01 [$-0.03$; 0.04], $P(\beta < 0) = 0.32$ fledging success: 0.013 [$-0.04$;

0.06], P($\beta < 0$) = 0.31) and was thus not further considered. The setting allowed us to assign the maternal morph for each egg ($N_{Total}$ = 962 eggs: 701 eggs laid by Independents, 196 eggs laid by Satellites and 65 eggs laid by Faeders). Exact maternal ID was known only for the subset of eggs that had developed ($N_{Developed}$ = 472 eggs: 374 eggs produced by 67 Independents, 66 eggs produced by 17 Satellites and 32 eggs produced by six Faeders) and for which we established parentage with genetic markers (see below). Females had access to males for at least 2 h daily. We collected eggs several times per day during daylight hours.

We weighed each egg to the nearest 0.01 g, marked them individually and incubated them at 37.5 °C and 55% humidity. We candled eggs every 4 days to confirm the progress of embryonic development. We opened eggs with stalled development and determined the age of the dead embryo based on Hamburger Hamilton embryonic developing stages from chicken Gallus gallus that have a similar incubation period[58]. We used eggs without any signs of development (HH index ≤ 1)[58] ($N_{not\ developed}$ = 130 eggs: 61 eggs laid by Independent, 48 eggs laid by Satellite and 21 eggs laid by Faeder females) to determine yolk steroid concentrations because hormones start to be metabolized by the embryo a few hours after incubation starts[59]. For offspring survival analyses, we took blood or tissue samples of chicks and embryos that developed for at least 2 days to determine their sex, morph and genetic mother with molecular markers described in ref. [29] and Supplementary Table 6. Hatched chicks were individually color-ringed and hand-raised with other chicks of similar age in heterosexual groups under ad libitum food until an age of 20 days, when they fledge[28,29]. Adult females of the breeding population were weighed to the nearest 0.1 g to calculate relative egg mass. We took these body weights in November or December of their first year to avoid effects of developing ovaries, and because ruffs store fat for their annual migration during summer/early autumn even in captivity.

Housing and rearing (permit #1232B-17) were approved by the Animal Care Committee of Simon Fraser University operating under guidelines from the Canadian Council on Animal Care.

**Genetic analyses**. Sex was identified using two sex-specific single nucleotide polymorphisms amplified with the Z43Bruffsex1 and RuffSexD7 primer sets[29]. Morphs were identified using a set of six diagnostic single nucleotide poly-morphism markers located in the inversion region[29]. We considered all adult females from the breeding colony alive in a given year as potential mothers. To determine the identity of the genetic mother, we genotyped samples of all potential mothers and offspring using 21 polymorphic microsatellites in multiplex reactions with up to eight microsatellite markers (details are provided in Supplementary Table 6). The mother ID was determined with parentage analysis according to ref. [60]. For the majority of offspring (92%), we identified unique mothers, who showed no mismatches across all genotyped loci. For the remainder, DNA degradation resulted in mismatches in parentage assignment as these were collected from dead embryos ($N$ = 12, all from Independent mothers). For these samples, we accepted as mothers females who were housed in the same pen as the egg was found and had the least number of mismatches ($3.6 \pm 2.6$ mismatches). When the embryo did not develop or died at an age of ≤ 2 days, we were unable to determine sex, morph and genetic mother. Consequently, these eggs were only used for analyses on laying rate and excluded from analyses on egg mass and from all survival analyses.

**Hormone analyses**. To determine hormone concentrations of testosterone, androstenedione and progesterone in yolk we sampled only eggs without any signs of embryonic development (HH index ≤ 1)[58] at first candling after 4 days. We were unable to obtain embryonic DNA, which would have been necessary to determine the mother ID. Consequently, some of these eggs may have been laid by the same female, leading to pseudoreplication. To minimize the impact of pseu-doreplication, we sampled only eggs belonging to the same pen when they were either laid on the same day, or eggs that were laid 14 days after the last sampling from the same pen, because ruffs typically produce four egg clutches with a minimal egg laying interval of 1 day[61] (and own observations). We weighed the yolk, added an equal amount of distilled water, homogenized it by shaking before storage at −20 °C. We analysed yolk samples at Max Planck Institute for Orni-thology (MPIO), Seewiesen (Germany) using a modification[62,63] of the method described in ref. [64] (details are provided in Supplementary Methods).

**Statistics**

*Laying rate*. To analyze whether female morphs have different laying rates, we used each pen's per capita laying rate as a response variable (i.e., the mean laying rate per female in a pen). The laying rate incorporates all eggs, including those with unknown mother ID. We used this approach because the proportion of infertile eggs differed across morphs ($X^2_{df=2}$ = 25.2, $p < 0.001$), meaning that restricting the analysis to the subset of developed eggs would have biased the results. We used a linear model to calculate posterior means and 95% CrI in relation to the maternal morph, including the sampling year and the mean age of females per pen (orthogonal linear and quadratic term) as independent variables. We simulated 10,000 values from the joint posterior distribution of the model parameters using the function sim of the package arm[65] and a flat prior distribution to obtain model estimates with R 4.0.0[66]. The model estimates with 95% CrI represent the mean

and the 2.5% and 97.5% quantiles of the simulated values. Instead of p-values, we provide the posterior probabilities for specific hypotheses, calculated as the pro-portion of simulated values that met the hypotheses. Probabilities higher than 0.975 or lower than 0.025 would be significant in frequentist statistics. We analyzed residual distributions graphically using normal quantile-quantile plots and by plotting residuals against leverage and against fitted values of each factor. We used the same residual analysis and the same simulation procedure to model effects of the other response variables (egg mass, yolk steroid concentrations and offspring survival). Female mean age per pen showed a quadratic relationship with laying rate, but the mean age distributions did not differ by morph (Independents: 6.2 years [5.8 −6.6], Satellites: 6.5 years [5.8 −7.2] and Faeders: 6.4 years [5.4 −7.5], Fig. 1).

*Egg mass*. To analyse whether the maternal morph influenced egg mass, we first used a linear model to calculate posterior means of egg mass and 95% CrI in relation to the maternal morph, with sampling year and mother ID included as random factors. Second, we prepared a linear model to calculate the posterior mean deviation of the observed egg mass from the expected egg mass (based on other shorebird species in ref. [41]) relative to the female's body weight: (observed egg mass/expected relative egg mass)-1. We calculated the expected egg mass using the formula egg mass = $0.613*$female body mass$^{0.726}$[41]. In both models, we used maternal morph as fixed factor, and year and mother ID as random factors[67]. As only six of 20 Faeder females produced fertile eggs, we present egg mass clustered by female ID for this morph (Supplementary Fig. 6).

*Yolk hormones*. We fitted three linear models with each of the hormone con-centrations as the response variable with female morph and sampling year as fixed effects. Since we were unable to determine mother ID, our results should be interpreted with caution. We log-transformed concentrations to normalize the residuals. We calculated posterior means of testosterone, androstenedione and progesterone and 95% CrI in relation to the maternal morph and sampling year[67].

*Offspring survival*. As the female morphs on average differ in the mass of the eggs they produce, we modeled offspring survival controlling for egg mass. We calcu-lated posterior means and 95% CrI of female morphs independent of egg mass, i.e., by setting egg mass to the overall mean. The variable "maternal morph" in these models hence gives the effect of female morph after controlling for variation in egg mass, which then mainly reflects differences in egg composition. We analysed offspring survival probability until hatching and fledging, two time points with developmental relevance. Ruffs start to fly at an age of approximately 20 days post hatching[28] and hence we defined fledging success as survival until this age. We used two different approaches. First, we calculated posterior means of hatching/fledging success and their 95% CrI to the fixed factors maternal morph, offspring sex, offspring morph and egg mass. We included sampling year and mother ID as random factors into the models, and used a binomial distribution for hatching and fledging probability. We measured hatching success from all eggs that showed signs of embryonic development, did not have any cracks when found and were not part of different experiments ($N$ = 400 eggs that were produced by 66 Independent, 15 Satellite and six Faeder females; egg number for each maternal morph is provided in Fig. 2, top left panel). Thus, our measure of hatching success excludes infertile eggs. We analyzed fledging success of all hatched chicks ($N$ = 165 chicks that had 49 Independent, nine Satellite and four Faeder mothers; chick number for each maternal morph is provided in Fig. 2, top right panel). From simulations, we obtained a total of 18 posterior distributions (i.e., maternal morph: three levels*embryo morph: three levels*embryo sex: two levels) for both hatching and fledging probability dependent on egg mass. To avoid overfitting the model, we did not include interactions among fixed factors. The results for all levels of embryo sex and morph with respect to egg mass are provided in Supplementary Fig. 3. To avoid redundancy, we visualize the effects of maternal morph on hatching/fledging success only for eggs with male Independent offspring and the overall mean egg mass (Fig. 2). Similarly, we only show the effect of egg mass on hatching/fledging success for eggs from Independent mothers with female Independent offspring (Fig. 2). For offspring from Faeder females, we present hatching success clustered by female ID (Supplementary Table 7).

For all fixed factors, we calculated the effect sizes through simulations of the posterior distributions. As the variables maternal and embryo morph have three levels, we modified the procedures in ref. [65] to obtain only a single average effect size. We first calculated the differences of simulated posterior distributions between morph comparisons. If the mean effect size between two morphs was negative, we multiplied the simulations by minus one. Based on these converted simulation values, we then calculated the average effect size of the three morphs and 95% CrI from the grand mean (and 95% CrI). For the effect size of offspring sex, we calculated the difference between males and females, and for egg mass, we multiplied the simulations of the posterior distribution by two times the standard deviation of egg mass[65,67].

Second, we modeled daily survival of embryos/chicks including all fixed factors that had shown an effect size that was different from zero using Cox models (R package 'survival'[68]). This enabled us to visualize morph-specific survival until fledging age and pinpoint critical developmental periods with survival differences graphically for all offspring that started embryonic development. For this analysis,

we used the same data sets as in the models for hatching/fledging success model, plus the 72 additional eggs from which embryos had been sampled as part of another experiment. For the survival analysis, these eggs were included as censored data points with unknown fate. Additionally, we modeled chick survival including egg mass as binary variable with two levels, low and high mass, which were separated by the overall mean of the egg mass distribution[67] (Supplementary Fig. 3, Supplementary Fig. 4).

*Models of male fertilization success required for polymorphism stability.* We explored whether and how higher fitness components for inversion morphs could compensate for the negative effects in inversion females. We calculated the necessary fitness modifications for three components, assuming stable allelic frequencies between generations: (1) reproductive success of males, (2) adult survival of females and (3) adult survival of males. We first calculated the allele frequencies of the supergene variants based on genetically determined morph frequencies of males and females sampled from a wild population[18], taking into account the established inversion region genotypes of the morphs[17,18]. For the reproductive success models, we then calculated the required fertilization success for Independent males. We set up three models. For the "Observed" model, we expressed the required proportion of eggs fertilized by Independent males as a function of the frequency of the Independent allele, the observed proportion of fledglings (Fig. 4a) produced by each female morph and the proportions of female morphs[18]. With this we solved for the required frequency of eggs fertilized by inversion males analogous to the required frequency of eggs fertilized by Independent males. Finally, we divided the required proportions of eggs fertilized by each male morph through the respective genetic male morph frequencies from the wild[18] to obtain estimates of the male fertilization success per capita.

For comparison, we modeled the required male fertilization success to maintain a stable polymorphism assuming no differences in reproductive success among female morphs, but still considering the homozygous lethality ("Equal output" model) and assuming that Faeder females' reproductive output would be zero ("No Faeder output" model). The sole difference between the models was the proportion of fledglings produced by each female morph. To evaluate the compensations through differences in adult survival alone, we determined the differences required in either male or female lifespan for the three morphs based on the calculated coefficients from the "Observed" model, assuming no compensation through male fertilization success. We expressed the changes for the inversion morphs relative to Independent lifespan. To obtain the fold-change in female lifespan, we divided the number of fledglings produced by Independent females through the respective number produced by either Satellites or Faeders. Conversely, for males, we divided the coefficients "male fertilization success per capita" of Satellites and Faeders through the respective coefficient of Independent males[67].

**Reporting summary**. Further information on research design is available in the Nature Research Reporting Summary linked to this article.

## Data availability

The datasets generated and analysed during the current study are available in Edmond the Open Research Data Repository of the Max Planck Society, https://doi.org/10.17617/3.71. Source Data are provided within the Source Data file. Source data are provided with this paper.

## Code availability

The code for the evolutionary model and all statistical analyses are available as R scripts in Edmond the Open Research Data Repository of the Max Planck Society, https://doi.org/10.17617/3.71.

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

## Acknowledgements

We thank Stephanie Rolio, Aaron Walchuk, Soma Marton and Gabi Trainor for help with animal care and technical assistance in sample collection, Monika Trappschuh, Sylvia Kuhn, Katrin Martin, Melanie Schneider and Rachel Tucker for help with laboratory work and parentage analyses and Fränzi Korner-Nievergelt for statistical assistance. Funding was provided by the Max Planck Society to C.K. and B.K.

## Author contributions

L.M.G.D., D.B.L., and C.K. designed and conceived the experiments, L.M.G.D., J.L.L., and D.B.L. collected the data, W.G., B.T., T.B., and B.K. contributed materials and analysis tools, L.M.G.D. analyzed the data and L.M.G.D. and C.K. wrote the manuscript with input by D.B.L. All authors revised and approved the final manuscript.

## Funding

## Competing interests

The authors declare no competing interests.
