## [Peer Review File · Nature Communications]

Intralocus conflicts associated with a supergeneREVIEWER COMMENTS

Reviewer #1 (Remarks to the Author):

Review for "Intralocus conflicts associated with a supergene," Nature Communications

Summary: In this manuscript, Lina Giraldi-Deck and colleagues evaluate the impact of male-morph supergene on female reproductive success in the ruff, *Calidris pugnax*. Three male morphs with distinct reproductive strategies are maintained in this species, where the single-locus chromosomal inversion that leads to the sneaker male (Faeder) phenotype is homozygous lethal, preventing fixation. While most work has focused on the behavior and mechanism of alternate male mating strategies in this species, the authors focus on female contributions to fitness by evaluating the impact of alternate supergene alleles on egg size, egg number, and fledging success. They find that the Faeder females produce fewer, smaller eggs with lower amounts of hatching and fledging. They also use the observed reproductive costs of the Faeder allele in females and known allele frequencies of the inversion in wild populations to model the increase in male reproductive success needed to maintain the Faeder allele despite the costs in females.

In general, I found this paper to be very interesting and well written. The system is nice to test questions of intralocus sexual conflict as it contains multiple morphs. The multi-year experiment to examine reproductive output is a nice way to evaluate fitness in a controlled manner. I'm very curious as to whether reduced female body and thus egg size explain the reduction in hatching success, or whether there are additional pleiotropic effects of the locus on development that modify maternal effect (seems like there could be some nice follow up experiments here). In my opinion, the largest strength of the paper is the connection of empirical female reproductive success and known allele frequencies to model the male success rate needed to maintain the allele. It's amazing that an allele with such a negative impact on female fitness can be maintained in the population.

- As a whole, the statistics in the paper are thoughtful and robust. They are well explained in the methods and throughout the text, which is particularly important because the differing sample sizes and asymmetry in reproductive success make the models very complicated. My one concern is that with only 4 fledgling Faeders, I'm not sure that there is sufficient power to detect impacts of inversion type on fledging. The authors find an effect of egg size on fledging – does this hold if the Faeder samples are removed?

- I would also like to have seen a little more context placed around the hormone findings in the discussion. Could androstenedione levels be responsible to low hatching success?

Reviewer #2 (Remarks to the Author):

This study aims to understand the selection regime underlying the balanced polymorphism in the ruff, and in the broader perspective how balanced polymorphisms in this and other similar systems are maintained over time. Ruffs mate in leks and have three distinct male plumage morphs (Independents, Satellites and Faeders) that are controlled by a c. 5 Mbp-long supergene with three alleles (I, S, F) that have been segregating as distinct haplotypes over long evolutionary timescales due to very low rate of recombination. S and F are dominant over I, and S and F are lethal in homozygous form (SS and FF) as well as in combination (SF). Thus, the morphs have the following genotypes: Independents are II, Satellites are IS and Faeders are IF. This is an autosomal locus and also females have these genotypes. As S and F carry a load, we do not expect them to increase to high frequencies, and to persist in the population we expect them to have benefits that compensate the cost. In the ruff system, this benefit is generally believed to be related to negative frequency dependent mating advantage in males: the morph that is presently occurring in low numbers gains a mating advantage over the other morphs during leks.

In the present paper, the authors propose, and provide data and results for, yet another fitness cost for one of the genotypes: IF-females are smaller and lay smaller eggs than II-females and IS-females. Since smaller eggs have reduced hatchability as shown in this paper, IF-females have lower fitness. An additional cost for the IF-genotype suggests that there might be some counter acting benefit, and they calculate the expected reproductive success that IF-males must obtain to uphold the allele frequencies in the population. It is concluded that there is an ongoing intra-locus sexual conflict at the morph locus acting on the F-allele (i.e. the F-allele has low fitness in females, and high fitness in males). This provides the novel perspective that there are other evolutionary processes acting on the balanced polymorphism in ruff in addition to negative frequency dependence selection on male mating success on leks.

The data comes from a captive population which is understandable because gathering such data in the wild would have been very difficult if not impossible. It is a unique experimental population that has produced a unique data set. Nevertheless, a main problem with this study is that the results are based on data of only 6 IF-females, which as I understand were kept in a single enclosure and are thus not independent (if true, this should be explicitly stated in methods). To make it clear that the results hold statistically despite being based on only 6 IF-females, I strongly suggest that the authors present results from frequentist statistics whenever possible, using female-ID as a factor in the models. The authors do mention "Probabilities higher than 0.975 or lower than 0.025 would be significant in frequentist statistics", and it would be appropriate to also present such data and results (this can be given as supplement results if not in the main text). The associations presented in Figure 1 and 3a should be possible to analyse with such models (frequentist statistics including female-ID), whereas I believe data used in Figure 2 is based on eggs with unassigned mothers, and then results from frequentist statistical models without including female-ID can be presented.

An uncertainty is whether the interpretations of the results are applicable to wild conditions. The negative frequency dependency in the system makes realized fitness highly dependent on allele frequencies and which morphs are present in any particular population/lek. The current results suggest that females carrying the more common alleles (I and S) are not affected by the fitness cost, meaning that the suggested sexual conflict might play a minor role for the maintenance of balanced polymorphism in this system overall. Also, the antagonism between males and females over the F-allele remains uncertain, as the current study does not show that IF-males are experiencing higher fitness than II- and IS-males. I believe that it remains to be understood how the suggested sexual antagonism over F relates to the cost associated with homozygosity (of F), and whether it contributes to lowering the frequency of F and IF-individuals in natural conditions. Also, could potentially the current result rather reflect a sexual conflict on body size (through egg size dependent hatchability) acting on all morphs, which happen to show up particularly strongly in IF-females as they are the smallest (and only 6 in the present study)? This can potentially be addressed by models including morph, body size and female-ID.

Reviewer #3 (Remarks to the Author):

Although supergenes underlie some of the most spectacular polymorphisms in nature we understand little about how they are maintained. In this manuscript Giraldo-Deck and colleagues examine the role of interlocus conflict in maintaining a supergene in the ruff. In this system, a supergene underlies 3 male reproductive morphs (independent, faeder, and satellite). Here, the authors examine the fitness consequences of morph on females to look for signature of intralocus sexual conflict (IASC). They find that faeder females had lower reproductive success than other morphs and conclude that due to this, higher reproductive success by faeder males is necessary to maintain the polymorphism.

The data and methodology are sound and the paper is well written and straightforward. However, I am not convinced that IASC in this system is helping to maintain the supergene polymorphism. The polymorphism can be lost via either fixation or loss of the faeder variant. As the inversion is

homozygous lethal there is no risk of fixation and the fitness consequences of the variant in females actually make the variant more likely to be lost. Furthermore, there are conflicting opinions as to whether IACS can actually help to maintain variation. While the findings presented here do provide a piece of the puzzle, I do not feel that the advance is large enough to warrant publication in Nature Communications.

Additional Notes:

A large amount of emphasis was placed on the model (e.g., the majority of figure 3) but I was not convinced that it adds much to the paper. Given that the female faeder suffers a fitness disadvantage it is clear that the males must have a fitness advantage at least in some scenarios. The model quantifies this but assumes random mating which is highly unlikely in this system. It would be helpful to add more information about mating dynamics in this system to the discussion. Other models of IASC should also be discussed briefly.

Line 202 - I disagree that the inversion here is a parasitic element. Broadly parasites can be defined as elements that do not convey a benefit to the host. However, the faeder variant is clearly beneficial in males especially when the morph is at a low frequency.

Figure 4. There is no feedback here between sexual antagonism and NFDS. As the faeder morph increases in frequency both IACS and NFDS will contribute to reduce faeder fitness.

REVIEWER COMMENTS

Reviewer #1 (Remarks to the Author):

Review for "Intralocus conflicts associated with a supergene," Nature Communications

Summary: In this manuscript, Lina Giral-di-Deck and colleagues evaluate the impact of male-morph supergene on female reproductive success in the ruff, *Calidris pugnax*. Three male morphs with distinct reproductive strategies are maintained in this species, where the single-locus chromosomal inversion that leads to the sneaker male (Faeder) phenotype is homozygous lethal, preventing fixation. While most work has focused on the behavior and mechanism of alternate male mating strategies in this species, the authors focus on female contributions to fitness by evaluating the impact of alternate supergene alleles on egg size, egg number, and fledging success. They find that the Faeder females produce fewer, smaller eggs with lower amounts of hatching and fledging. They also use the observed reproductive costs of the Faeder allele in females and known allele frequencies of the inversion in wild populations to model the increase in male reproductive success needed to maintain the Faeder allele despite the costs in females.

In general, I found this paper to be very interesting and well written. The system is nice to test questions of intralocus sexual conflict as it contains multiple morphs. The multi-year experiment to examine reproductive output is a nice way to evaluate fitness in a controlled manner. I'm very curious as to whether reduced female body and thus egg size explain the reduction in hatching success, or whether there are additional pleiotropic effects of the locus on development that modify maternal effect (seems like there could be some nice follow up experiments here). In my opinion, the largest strength of the paper is the connection of empirical female reproductive success and known allele frequencies to model the male success rate needed to maintain the allele. It's amazing that an allele with such a negative impact on female fitness can be maintained in the population.

We appreciate this positive assessment of our work!

Regarding your question about egg size explaining the reduced hatching success: it did not. In our models we statistically controlled for egg size (=egg mass, line 94). We have changed the text to make this clearer, i.e. that the reduced hatching success observed in Faeder females is not a consequence of those females laying smaller eggs (lines 94-96). Additionally, we clarified in Fig. 2 that the presented effects of maternal morph on hatching and fledging success were calculated using the overall mean egg mass (hatching: 18.61g, fledging: 18.68g). The estimates of lower or higher egg mass are presented in Supplementary Fig. S3. The reduced embryonic survival of eggs laid by Faeder females thus points indeed towards additional pleiotropic effects of the maternal Faeder allele on embryonic survival.

- As a whole, the statistics in the paper are thoughtful and robust. They are well explained in the methods and throughout the text, which is particularly important because the differing sample sizes and asymmetry in reproductive success make the models very complicated. My one concern is that with only 4 fledgling Faeders, I'm not sure that there is sufficient power to detect impacts of inversion type on

fledging. The authors find an effect of egg size on fledging – does this hold if the Faeder samples are removed?

Indeed, we did not detect an effect of maternal morph on fledging success and this could be the result of having too few hatched offspring of Faeder mothers in our sample. However, the general effect of egg mass on fledging success that we report is clear and holds when we remove the Faeder samples (Fig. R1, Table R1), the estimated relationships are nearly identical. When analyzing egg mass, our model controls for maternal morph.

Fig. R1. Influence of egg mass on fledging probability (mean \pm 95% CrI) according to the model that includes chicks from Faeder mothers (used in the manuscript; continuous grey line and grey shading) and excluding chicks from Faeder mothers (broken green line and green shading). Jittered orange dots indicate egg mass and survival of the four chicks that hatched from Faeder mothers.

Table R1. Posterior distributions of parameter estimates on fledging probability with eggs from Faeder mothers (model used in the manuscript) and without eggs from Faeder mothers. Both models include mother ID and sampling year as random factors. The intercept provides the estimate for Independent mothers for a female, Independent offspring and a mean egg mass. Given are mean values, 95% CrI, and the posterior probability of the hypothesis that the parameter is smaller than zero. P values in bold are comparable to a significant effect ($P < 0.05$) according to frequentist statistics.

Parameter	Fledging success with eggs from Faeders ($N_{\text{Offspring}}=165$, $N_{\text{Mothers}}=62$)			Fledging success without eggs from Faeders ($N_{\text{Offspring}}=161$, $N_{\text{Mothers}}=58$)		
	Mean	95% CrI	P ($\beta < 0$)	Mean	95% CrI	P ($\beta < 0$)
Intercept	0.85	0.18; 1.51	0.01	0.87	0.22; 1.53	<0.01
Satellite mother	0.32	-0.93; 1.58	0.31	0.32	-0.96; 1.59	0.31
Faeder mother	0.20	-2.19; 2.58	0.43	-	-	-
male offspring	-0.50	-1.2; 0.23	0.91	-0.49	-1.21; 0.22	0.91
Satellite offspring	0.23	-0.6; 1.05	0.29	0.23	-0.59; 1.05	0.30

Faeder offspring	0.24	-0.77; 1.26	0.32	0.25	-0.76; 1.29	0.32
egg mass	0.50	0.12; 0.89	0.0047	0.47	0.09; 0.84	0.0075

- I would also like to have seen a little more context placed around the hormone findings in the discussion. Could androstenedione levels be responsible to low hatching success?

Good point, we have extended the discussion in this regard. Please see lines 175-190.

Reviewer #2 (Remarks to the Author):

This study aims to understand the selection regime underlying the balanced polymorphism in the ruff, and in the broader perspective how balanced polymorphisms in this and other similar systems are maintained over time. Ruffs mate in leks and have three distinct male plumage morphs (Independents, Satellites and Faeders) that are controlled by a c. 5 Mbp-long supergene with three alleles (I, S, F) that have been segregating as distinct haplotypes over long evolutionary timescales due to very low rate of recombination. S and F are dominant over I, and S and F are lethal in homozygous form (SS and FF) as well as in combination (SF). Thus, the morphs have the following genotypes: Independents are II, Satellites are IS and Faeders are IF. This is an autosomal locus and also females have these genotypes. As S and F carry a load, we do not expect them to increase to high frequencies, and to persist in the population we expect them to have benefits that compensate the cost. In the ruff system, this benefit is generally believed to be related to negative frequency dependent mating advantage in males: the morph that is presently occurring in low numbers gains a mating advantage over the other morphs during leks.

In the present paper, the authors propose, and provide data and results for, yet another fitness cost for one of the genotypes: IF-females are smaller and lay smaller eggs than II-females and IS-females. Since smaller eggs have reduced hatchability as shown in this paper, IF-females have lower fitness.

Please note that the reduced hatchability of eggs from Faeder mothers was independent from egg mass (lines 94-96, see also our reply to reviewer 1). This means that the Faeder allele has several negative effects on the reproductive success of Faeder females. 1. Faeder females laid fewer eggs than Independent or Satellite females 2. Their offspring had a reduced hatchability, probably due to variation in egg composition (see discussion lines 175-190) 3. Faeder females laid smaller eggs and survival of hatched chicks increased with egg mass. All these effects are additive and result in an overall strongly reduced reproductive success for Faeder females in comparison with the other two morphs (summarized in Figure 3a).

An additional cost for the IF-genotype suggests that there might be some counter acting benefit, and they calculate the expected reproductive success that IF-males must obtain to uphold the allele

frequencies in the population. It is concluded that there is an ongoing intra-locus sexual conflict at the morph locus acting on the F-allele (i.e. the F-allele has low fitness in females, and high fitness in males). This provides the novel perspective that there are other evolutionary processes acting on the balanced polymorphism in ruff in addition to negative frequency dependence selection on male mating success on leks.

The data comes from a captive population which is understandable because gathering such data in the wild would have been very difficult if not impossible. It is a unique experimental population that has produced a unique data set. Nevertheless, a main problem with this study is that the results are based on data of only 6 IF-females, which as I understand were kept in a single enclosure and are thus not independent (if true, this should be explicitly stated in methods). To make it clear that the results hold statistically despite being based on only 6 IF-females, I strongly suggest that the authors present results from frequentist statistics whenever possible, using female-ID as a factor in the models. The authors do mention "Probabilities higher than 0.975 or lower than 0.025 would be significant in frequentist statistics", and it would be appropriate to also present such data and results (this can be given as supplement results if not in the main text). The associations presented in Figure 1 and 3a should be possible to analyse with such models (frequentist statistics including female-ID), whereas I believe data used in Figure 2 is based on eggs with unassigned mothers, and then results from frequentist statistical models without including female-ID can be presented.

We have clarified that the data on the reproductive success actually came from three pens during the three years of study period (lines 246-248). We studied the reproductive success of 20 Faeder females (line 78), which laid 65 eggs during this time (line 253) meaning that we had a reasonable sample size. Faeder females had a lower annual laying rate than Independent or Satellite females (lines 79-80; Fig. S1, table S1). The reviewer is correct that for some analyses, i.e. the hatching and fledging success, we could only analyze the reproductive success of a subset of these females though. For example, only six females produced offspring that survived at least two days yielding enough DNA for parentage analyses. But this is rather a direct consequence of the fact that Faeder females had low fecundity and their eggs did not develop very well, which is a major result of our study.

We appreciate that this was not expressed clearly enough. Therefore, we have clarified that models on egg mass and offspring survival had female ID included as random factor (lines 646-647, 361, 619). Please note that the negative effects of the Faeder allele on female reproduction were statistically clear, despite their low sample size. To be fully transparent, we have now included raw data of eggs from Faeder mothers clustered by Faeder female ID (lines 340-341 and lines 377-378; Supplementary table S7, Fig. S6) to show that the observed reduced egg mass and reduced hatching success were not driven by individual Faeder females but rather this is a consistent trait of all Faeder females studied.

To address the reviewer's concern we provide results from frequentist statistics for review only (table R2-R5). As you will see, the results are fully consistent with our Bayesian models. However, we would prefer not to add these to the manuscript as the use of Bayesian statistics is most appropriate in our case and the frequentist models do not add much more. We used generalized mixed models to evaluate hatching and fledging success with Bayesian statistics for the following reasons. First, ecological statisticians argue, that Bayesian methods are the only exact way to draw inference from generalized mixed models (Korner-Nievergelt 2015; Bolker et al. 2008). These models allowed us to include female ID and sampling year as random factors (Fig. 2). As we wanted to be consistent with all

our analyses we used Bayesian methods throughout this study. Second, Bayesian methods make the estimation of credible intervals for combined models straight forward, as these are based on the simulated values of the posterior distribution. In our study the use of the posterior distributions of annual laying rates, hatching success and fledging success allowed us to model the combined reproductive output for each female morph (Fig. 3a) and to infer male reproductive success needed for stable allelic frequencies (Fig. 3b). Third, Bayesian statistics allowed us to obtain the posterior probabilities for specific hypotheses, instead of having to rely on generic null hypothesis testing, which has been criticized for many years (e.g. Anderson et al. 2000; Matthews 2000; Ridley et al. 2007; Burnham & Anderson 2014).

To facilitate the interpretation of Bayesian statistics for unfamiliar readers we stated: “Instead of *p*-values, we provide the posterior probabilities for specific hypotheses, calculated as the proportion of simulated values that met the hypotheses. Probabilities higher than 0.975 or lower than 0.025 would be significant in frequentist statistics.” (lines 321-323). In addition, tables containing statistical information (Table S1, S2, S3, S4) have bold *P* ($\beta < 0$) values when they would have been comparable to a significant effect ($P < 0.05$) according to frequentist statistics.

Our strong preference would be to report only the results of the Bayesian Models. However, if the editor feels different about this we would be willing to add frequentist model results to the supplementary material.

Table R2 Results of a linear mixed model examining the effect of the female morph on egg mass with female ID and sampling year included as random factors. The between-female standard deviation was 0.98, the between-sampling year standard deviation was 0.18 and the residual standard deviation was 0.95. *P* values in bold are considered significant ($P < 0.05$).

Fixed effects	Estimate	SE	P value
Intercept	18.81	0.17	<0.001
Satellite mother	-0.10	0.31	0.756
Faeder mother	-1.53	0.48	0.002

Table R3 Results of a linear mixed model examining the effect of the female morph on deviation of the observed egg mass from the expected egg mass according to female body mass. For expected egg mass calculations we used the formula egg mass = 0.613*female body mass^{0.726}. We included female ID and sampling year as random factors in the model. The between-female standard deviation was 0.11, the between-sampling year standard deviation was 0.01 and the residual standard deviation was 0.06. *P* values in bold are considered significant ($P < 0.05$).

Fixed effects	Estimate	SE	P value
Intercept	0.06	0.02	<0.001
Satellite mother	0.01	0.04	0.851
Faeder mother	0.20	0.05	<0.001

Table R4 Results of a generalized mixed model examining the effects of maternal morph offspring sex, offspring morph and egg mass on hatching success (0=unhatched; 1=hatched). We included female ID and sampling year in the model. The between-mother ID standard deviation was 0.84 and the between-sampling year standard deviation was 0.10. Variance of inflation factor was lower than 1.2 in all predictors. *P* values in bold are considered significant ($P < 0.05$).

Fixed effects	Estimate	SE	P value
---------------	----------	----	----------------

Intercept	-0.49	0.24	0.046
Satellite mother	-0.44	0.46	0.333
Faeder mother	-1.60	0.75	0.032
male offspring	0.31	0.24	0.194
Satellite offspring	0.22	0.28	0.427
Faeder offspring	0.19	0.35	0.587
egg mass	0.04	0.14	0.771

Table R5 Results of a generalized mixed model examining the effects of maternal morph offspring sex, offspring morph and egg mass on fledging success (0=unfledged; 1=fledged). We included female ID and sampling year in the model. The between-mother ID standard deviation was 0.51 and the between-sampling year standard deviation was <0.001. Variance of inflation factor was lower than 1.3 in all predictors. *P* values in bold are considered significant ($P < 0.05$).

Fixed effects	Estimate	SE	P value
Intercept	0.85	0.36	0.019
Satellite mother	0.32	0.66	0.624
Faeder mother	0.20	1.22	0.873
male offspring	-0.50	0.38	0.192
Satellite offspring	0.23	0.43	0.594
Faeder offspring	0.24	0.53	0.643
egg mass	0.50	0.21	0.014

An uncertainty is whether the interpretations of the results are applicable to wild conditions. The negative frequency dependency in the system makes realized fitness highly dependent on allele frequencies and which morphs are present in any particular population/lek. The current results suggest that females carrying the more common alleles (I and S) are not affected by the fitness cost, meaning that the suggested sexual conflict might play a minor role for the maintenance of balanced polymorphism in this system overall. Also, the antagonism between males and females over the F-allele remains uncertain, as the current study does not show that IF-males are experiencing higher fitness than II- and IS-males.

Indeed, measuring the female fitness in captivity is a drawback of our study. However, as this reviewer stated before a similar study would not be feasible in the wild. Breeding Faeders (both males and females) are very rare in the wild (~1-2%) and wild Faeder males are cryptic and highly mobile. Therefore, it would be a major challenge even for a long-term study to collect comparable fitness data in the wild. However, the morph frequencies seem stable over time and space, most likely because the cosmopolitan ruffs are highly panmictic. We have clarified the text in this regard (lines 113-114).

As the reviewer points out, Faeders represent only a small part of the ruff population. However, regardless of its low ("minor") frequency, the Faeder allele encodes an entire behavioural strategy. Both, Faeder allele and strategy are vital components of the genetic and behavioural polymorphism in this species. Because the Faeder allele is dominant, it will be only transmitted to the next generation

by Faeder individuals. It follows that its persistence depends on the combined reproductive fitness of Faeder males and Faeder females, which we examined in our study.

To measure the reproductive advantage of males in our captive population was not feasible as the males are familiar with each other and recognize Faeder males as competitors. Faeder males employ a sneaking strategy and these behavioural strategies are typically under strong negative frequency dependent selection. In our population, the Faeder males occur in a much higher frequency than in the wild and their mobility is constrained. Therefore, we rather used a set of analytical evolutionary models to identify how the observed reproductive burden in Faeder females could be compensated to maintain allelic frequencies in equilibrium. Our results suggest that full compensation through morph-biased survivorship alone is unrealistic (lines 128-131). Rather, the detrimental effects of the Faeder allele on female reproduction are most likely maintained by higher-than-average reproductive success in males (lines 144-147).

Our study suggests that a combination of sexual antagonism and negative frequency dependent selection (Fig. 4) is a plausible explanation for maintaining the Faeder allele at the observed low frequency in the wild.

I believe that it remains to be understood how the suggested sexual antagonism over F relates to the cost associated with homozygosity (of F), and whether it contributes to lowering the frequency of F and IF-individuals in natural conditions.

We are unclear what the reviewer means with this comment. We do not say that that sexual antagonism over F is related to homozygous lethality of F (or S). We have changed the text to make this clearer (lines 151-153). The lethality of F when homozygous is only a small factor for the poor reproductive success of Faeder females (see Fig. 3a, grey squares at egg stage). However, based on the empirical data on female reproductive success (summarized in Fig. 3a) and the frequency distribution of the morphs in the wild (in Figure 3c), it is clear that Faeder females produce so few Faeder offspring that the Faeder allele cannot be maintained in the population unless there is a compensation mechanism. Our evolutionary models suggest that the most likely mechanism is higher than average reproductive success by Faeder males that maintains the Faeder allele in the populations. In fact, most of the Faeder offspring are raised by Independent mothers who mated with Faeder males (Fig. 3c).

Also, could potentially the current result rather reflect a sexual conflict on body size (through egg size dependent hatchability) acting on all morphs, which happen to show up particularly strongly in IF-females as they are the smallest (and only 6 in the present study)? This can potentially be addressed by models including morph, body size and female-ID.

Good point. As we stated before, the Faeder allele has pleiotropic effects on female reproduction. Body size is certainly one trait affected by the inversion – Faeder females are smaller (line 19) and could have influenced Faeder female reproduction. We have checked this but found that there is no statistically clear effect (table R6 and table R7). Including adult body mass as predictor for egg mass, suggests that low egg mass in Faeder females is mainly driven by morph and not by a lower body mass compared to Independents or Satellites. Please note that adult body mass refers to body weights

during their first winter and not prior to laying. But this is clearly a fruitful avenue for further research. Thus, we added to the discussion that to understand the proximate associations between the supergene variants and female reproduction, studies on how the supergene variants effect female physiology and morphology would be needed (lines 187-190).

Table R6. Posterior distributions of parameter estimates for female morph and female body mass on egg mass (in g) with female ID and sampling year included as random factors in the model. The intercept provides the estimate for Independent females. Given are the mean, 95% CrI, and the posterior probability of the hypothesis that the parameter is smaller than zero. The between-female standard deviation was 1.00, the between-sampling year standard deviation was 0.04 and the residual standard deviation was 0.91. P values in bold are comparable to a significant effect ($P < 0.05$) according to frequentist statistics.

parameter	mean	95% CrI	P ($\beta < 0$)
Intercept	19.51	17.41; 21.52	<0.001
Satellite mother	-0.20	-0.85; 0.48	0.719
Faeder mother	-1.56	-2.75; -0.36	0.994
adult body mass	-0.01	-0.03; 0.013	0.755

Table R7 Frequentist results of a linear mixed model examining the effect of the female morph and female body mass on egg mass with female ID and sampling year included as random factors. The between-female standard deviation was 1.00, the between-sampling year standard deviation was 0.04 and the residual standard deviation was 0.91. P values in bold are considered significant ($P < 0.05$).

Fixed effects	Estimate	SE	P value
Intercept	19.51	1.05	<0.001
Satellite mother	-0.20	0.34	0.566
Faeder mother	-1.56	0.61	0.013
adult body mass	-0.01	0.01	0.489

Reviewer #3 (Remarks to the Author):

Although supergenes underlie some of the most spectacular polymorphisms in nature we understand little about how they are maintained. In this manuscript Giraldo-Deck and colleagues examine the role of interlocus conflict in maintaining a supergene in the ruff. In this system, a supergene underlies 3 male reproductive morphs (independent, faeder, and satellite). Here, the authors examine the fitness consequences of morph on females to look for signature of intralocus sexual conflict (IASC). They find that faeder females had lower reproductive success than other morphs and conclude that due to this, higher reproductive success by faeder males is necessary to maintain the polymorphism.

The data and methodology are sound and the paper is well written and straightforward. However, I am not convinced that IASC in this system is helping to maintain the supergene polymorphism. The polymorphism can be lost via either fixation or loss of the faeder variant. As the inversion is homozygous lethal there is no risk of fixation and the fitness consequences of the variant in females actually make the variant more likely to be lost. Furthermore, there are conflicting opinions as to whether IASC can

actually help to maintain variation. While the findings presented here do provide a piece of the puzzle, I do not feel that the advance is large enough to warrant publication in Nature Communications.

We appreciate this comment. While we respectfully disagree with the reviewer here, we have clarified and qualified our view. We demonstrate clear detrimental effects of the Faeder haplotype on female reproductive success. The Faeder allele has not been lost, but remains at stable low frequencies, despite its genetic handicap of homozygous lethality and its reproductive burden for females. The question then is: If the Faeder variant is such a handicap, why did it not go extinct but has survived already for such a long time? We did not say that IASC alone will maintain the morph variation although a new model that we now cite suggests that this could be the case in species with alternative reproductive tactics (lines 214-215, reference 51 (Gamble & Calsbeek 2021)). Rather based on the outcome of our evolutionary models, we argue that balancing selection through the *combined effects* of IASC and negative frequency dependent selection could maintain the Faeder allele.

Thus we accept that this may not have been stated clearly enough before. Therefore, we have added to the revised text the vital details, i.e. that the Faeder inversion has existed already for approximately 3.8 million years (line 58) and that morph frequencies in wild populations appear generally stable over time and space (lines 113-114), and qualified our interpretation (line 221).

Please note that our discussion about IASC contributing to the maintenance of variation is backed up by a broad theoretical and empirical number of studies with nine references (ref. 8, 9, 10, 11, 12, 47, 50, 51, 52) included in the respective paragraph (lines 214-225).

Additional Notes:

A large amount of emphasis was placed on the model (e.g., the majority of figure 3) but I was not convinced that it adds much to the paper. Given that the female faeder suffers a fitness disadvantage it is clear that the males must have a fitness advantage at least in some scenarios.

We side with reviewer 1, who found great value in modelling the specifics and quantifying the required advantage for the males. Modelling the required male mating success for allelic stability allowed us to infer that variation in survival is highly unlikely to compensate for the negative effects on Faeder female reproduction (lines 128-131). The predicted increase of mating success in Faeder males suggests that sexual antagonism and negative-frequency dependent selection really are key to understanding how the Faeder allele can persist in Ruff populations, and provides testable predictions about Faeder male fertilization success.

The model quantifies this but assumes random mating which is highly unlikely in this system. It would be helpful to add more information about mating dynamics in this system to the discussion.

Indeed, our model assumes random matings between the morphs. There is no evidence in ruffs to suggest that this assumption is invalid. In captivity, matings between all morphs occur and we do not see a discrimination of inversion females against inversion males, rather over the years we have raised multiple offspring from such crosses (unpublished observations).

Other models of IASC should also be discussed briefly.

We have broadened the discussion as suggested and discuss one more model on IASC in alternative reproductive tactics that has been recently published (ref. 51: Gamble & Calsbeek 2021). In addition, we have included a review about how sexual conflict can maintain genetic diversity through balancing selection (ref. 52: Mank 2017).

Line 202 - I disagree that the inversion here is a parasitic element. Broadly parasites can be defined as elements that do not convey a benefit to the host. However, the faeder variant is clearly beneficial in males especially when the morph is at a low frequency.

We take your point about the benefits for Faeder males and have re-phrased the sentence (lines 226-227). We did not wish to give the impression that the Faeder allele is parasitic to the bearers. However, please note that we had not stated that the Faeder allele “is” a parasitic element but rather wrote that the inversion “resembles a parasitic element that is unable to persist without the ancestral haplotype”. This does hold true and we outline in the following sentences in this paragraph our arguments: 1) genetically: the Faeder allele cannot survive on its own as it is homozygous lethal, it always requires an ancestral allele, 2) behaviourally: the Faeder allele is maintained by male Faeders who feed of the lekking efforts of Independent males and 3) life-history: the vast majority of Faeders are reared by Independent mothers.

Figure 4. There is no feedback here between sexual antagonism and NFDS. As the faeder morph increases in frequency both IACS and NFDS will contribute to reduce faeder fitness.

Thanks for pointing this out. We have replaced “feedback” by “relationship”.

REVIEWER COMMENTS

Reviewer #1 (Remarks to the Author):

The authors thoroughly addressed my previous comments, and I found that the additions to the manuscript added both context and clarity. Thank you for adding the additional androstenedione context--very interesting!

Reviewer #2 (Remarks to the Author):

The response to the reviewers' comments and the edits to the MS have clarified my previous concerns. I have no further comments.

Reviewer #3 (Remarks to the Author):

I previously reviewed the article by Giraldo-Deck and colleagues. I still find it to be a nice data set and well-written paper. However, I am unconvinced that the findings have broad implications outside of the Ruff system.

The authors frame the manuscript around the questions of supergene evolution and maintenance ("These findings suggest that intralocus conflicts may play a major role in the evolution and maintenance of supergene variants." - line 27). However, the findings of the manuscript do not tell us how the Ruff supergene is maintained. The faeder variant is already known to be homozygous lethal and here the authors have shown yet another cost of this variant, i.e., reduced female reproductive output. They speculate that increased male fitness must then be responsible for the maintenance of the polymorphism but do not have any data to back this up. In fact, without showing a benefit in males, we cannot even call this intralocus conflict. In the end, the paper only shows that there is an additional female cost. While this is an important finding for the Ruff system, I cannot see how this is broadly applicable to other supergene systems or theoretical investigations of supergene maintenance.

While I do not think the article is suitable for Nature Communications I want to stress that this is a good data set and one that took time and care to put together. I appreciate the author's careful revision and look forward to seeing this published.

We are happy that our revision has been provisionally accepted and served to clarify previous concerns of reviewers one and two. We appreciate the opinion of reviewer three on our manuscript.

Reviewer #3

I previously reviewed the article by Giraldo-Deck and colleagues. I still find it to be a nice data set and well-written paper. However, I am unconvinced that the findings have broad implications outside of the Ruff system.

The authors frame the manuscript around the questions of supergene evolution and maintenance (“These findings suggest that intralocus conflicts may play a major role in the evolution and maintenance of supergene variants.” - line 27). However, the findings of the manuscript do not tell us how the Ruff supergene is maintained. The faeder variant is already known to be homozygous lethal and here the authors have shown yet another cost of this variant, i.e., reduced female reproductive output. They speculate that increased male fitness must then be responsible for the maintenance of the polymorphism but do not have any data to back this up. In fact, without showing a benefit in males, we cannot even call this intralocus conflict. In the end, the paper only shows that there is an additional female cost. While this is an important finding for the Ruff system, I cannot see how this is broadly applicable to other supergene systems or theoretical investigations of supergene maintenance.

While I do not think the article is suitable for Nature Communications I want to stress that this is a good data set and one that took time and care to put together. I appreciate the author’s careful revision and look forward to seeing this published.

We appreciate the criticism of this reviewer although we respectfully disagree. It is true that we did not measure the male fertilization success empirically. However, we did not “speculate that increased male fitness must then be responsible”, rather we modelled possible compensation mechanisms. Increased reproductive success by Faeder males was the mostly likely outcome. Compensation by longer survival alone would require completely unrealistic life expectancy differences between the morphs. Therefore, we maintain that the Faeder allele has a strong sexually antagonistic effect. This represents a strong intralocus conflict that contributes to the persistence of this genetic and behavioural polymorphism.